# Relationship of Multimorbidity, Obesity Status, and Grip Strength among Older Adults in Taiwan

**DOI:** 10.3390/ijerph18147540

**Published:** 2021-07-15

**Authors:** Ming-Hsun Lin, Chun-Yung Chang, Der-Min Wu, Chieh-Hua Lu, Che-Chun Kuo, Nain-Feng Chu

**Affiliations:** 1National Defense Medical Center, Division of Endocrinology and Metabolism, Department of Internal Medicine, Tri-Service General Hospital, Taipei 114, Taiwan; tim6801@msn.com (M.-H.L.); alpha1320@gmail.com (C.-Y.C.); undeca2001@gmail.com (C.-H.L.); 2Department of Internal Medicine, Kaohsiung Armed Forces General Hospital, Kaohsiung City 802, Taiwan; 3National Defense Medical Center, School of Public Health, Taipei 114, Taiwan; wangsn68@gmail.com; 4Department of Internal Medicine, Taoyuan Armed Forces General Hospital, Taoyuan City 325, Taiwan; lg1001010@gmail.com; 5Department of Internal Medicine, Tri-Service General Hospital, Taipei 114, Taiwan

**Keywords:** disease status, weight status, grip strength, older adults

## Abstract

**Background**: The combination of multiple disease statuses, muscle weakness, and sarcopenia among older adults is an important public health concern, and a health burden worldwide. This study evaluates the association between chronic disease statuses, obesity, and grip strength (GS) among older adults in Taiwan. **Methods**: A community-based survey was conducted every 3 years among older adults over age 65, living in Chiayi County, Taiwan. Demographic data and several diseases statuses, such as diabetes mellitus, hypertension, cerebrovascular disease, cardiovascular disease, and certain cancers, were collected using a questionnaire. Anthropometric characteristics were measured using standard methods. Grip strength was measured using a digital dynamometer (TKK5101) method. **Results**: A total of 3739 older individuals were recruited (1600 males and 2139 females) with the mean age of 72.9 years. The mean GS was 32.8 ± 7.1 kg for males and 21.6 ± 4.8 kg for females. GS significantly decreased most in males with cerebrovascular disease (from 33.0—29.5 kg, *p* < 0.001) and in females with diabetes mellitus (from 21.8—21.0 kg, *p* < 0.01). GS was highest in older adults with obesity (body mass index ≥ 27 kg/m^2^); however, there was no significant change of GS as the disease number increased. **Conclusion**: Older adults who have two, rather than one or greater than three chronic diseases, have significantly lower GSs than those who are healthy. Stroke and CKD for males, and hypertension and diabetes for females, are important chronic diseases that are significantly associated with GS. Furthermore, being overweight may be a protective factor for GS in older adults of both sexes.

## 1. Introduction

Multimorbidity is commonly defined as the coexistence of two or more adverse health conditions or diseases [1] and is an important issue among the ageing population. Its prevalence is approximately one in four adults below age 65 and increases as age increases. In the United States, three in four adults over age 65 have multimorbidity [2,3]. In Taiwan, the prevalence of multimorbidity was around four in five people above 80 [4]. Meanwhile, the ageing population increased significantly because of fertility decline and added residual life. The condition of multimorbidity also grows with ageing, and it is associated with an increased burden on healthcare costs [4,5]. Multimorbidity among older adults indicates a serious and crucial challenge not only for patient care but also for the healthcare systems in modern countries.

For older adults, progressive physiological muscle weakness not only contributes to age-related reductions in muscle capacity but also to declining neural activation [6]; muscle function and balance are more important than muscle mass [7]. Measurement of grip strength (GS) is a simple, quantifiable, and portable method used for evaluating the physical performance and nutritional status of older adults [8]. GS not only indicates sarcopenia, but is also a convenient, objective goal to assess physical performance, and an indispensable biomarker for all cause- and disease-specific mortality for older adults [9]. The aetiologies of sarcopenia include multiple complex factors, such as age-related hormone reduction, declined physical activity, and possible changes in the metabolic system (such as dysregulation of glucose metabolism [10] and the autonomic nervous system [11]). Apart from the age-related physiologic decline, multimorbidity is also important. In the past, the impact of GS on disease was usually discussed with a single chronic disease, such as cardiovascular disease (CVD), diabetes mellitus (DM), cerebrovascular disease (CVA), chronic kidney disease, or hypertension. Several recent studies investigated the impact of GS on multimorbidity and reported an inverse association between them [12,13,14,15,16,17].

The purpose of this study was to evaluate the association between different disease statuses and grip strength, as well as possible potential sex differences. We also examined the change in grip strength in multiple disease statuses and different weight statuses among elder adults in Taiwan. We also tried to identify GS in earlier stages and prevent the condition from worsening if people have other chronic diseases.

## 2. Materials and Methods

### 2.1. Study Population

From 2017 to 2019, we conducted a series of community-based health surveys among people age 65 or older, living in Chiayi County, Taiwan. The survey was conducted in each county every 3 years. The inclusion criteria were adults, 65–85 years of age, and the absence of infection or acute disorders within the last 3 weeks.

### 2.2. Questionnaire

Data on general demographics and lifestyle patterns (such as dietary pattern, status of smoking, and alcohol intake) were collected using a standard structured questionnaire. Disease statuses, such as diabetes mellitus, hypertension, CVD, CVA, chronic kidney disease, hyperlipidaemia, and certain cancers, were also obtained from the questionnaire and medical records. The participants were asked the following questions: “Did a doctor tell you that you have a disease such as [name of the disease]? Have you regularly been taking medications for chronic disease?”

### 2.3. Anthropometric Measurements

Anthropometric characteristics were measured using standard methods. Using a standard beam balance scale, we measured the body weight (BW) of participants, who were barefoot and wearing light indoor clothing, to an accuracy of 0.1 kg. Body height (BH) was also recorded to the nearest 0.5 cm using a stadiometer. We calculated the BMI as BW (kg) divided by the square of height (m^2^) and classified the BMI into the following four categories: normal weight (18.5–24 kg/m^2^), overweight (24–26.9 kg/m^2^), and obese (≥27 kg/m^2^), according to the definition of the Health Promotion Administration in Taiwan.

### 2.4. Grip Strength Measurement

Grip strength was measured with a digital hand dynamometer (TKK 5101Grip D; Takey, Tokio, Japan), which is a tool with an adjustable grip span, ranging from 3.5–7 cm and weighing from 5–100 kg [18]. Its precision is 0.1 kg All participants were seated following the procedure described by España-Romero et al. [19], with their elbows fully extended. Then, we measured the GS of either the right or left hand after 2–3 min of resting. We recorded two GS measurements and calculated the mean value for the analyses.

### 2.5. Approval by the Institutional Review Board (IRB)

All participants provided written informed consent as well as all information related to the study, and agreed to provide their general demographic data, questionnaire answers, anthropometric data, and blood samples. The IRB of Tri-service General Hospital approved this study (Approval No.: TSGHIRB-1-108-05-073).

### 2.6. Statistical Methods

All statistical data were analyzed using SPSS version 22 (IBM Corporation, New York, NY, USA). Continuous variables such as anthropometric measures and GS were described by sample means and SDs. The differences between groups were compared by student *t*-tests. For comparing subgroups and more than three groups, we used the analysis of variance (ANOVA) test and post hoc test. Meanwhile, categorical variables were described by number and percentage. We used chi-squared tests when comparing the differences between two groups, and when comparing more than two groups. Multiple regression analysis was applied to evaluate the relationship between grip strength and chronic disease status. A two-tailed *p* value of less than 0.05 was considered statistically significant. Box plots of GS among different disease statuses were drawn as illustrated in Figure 1.

## 3. Results

We recruited 3739 older respondents (1600 males and 2139 females), with a mean age of 76 years (ranging from 65 to 85 years). Table 1 presents the general characteristics of the participants. The mean age and GS were 72.9 ± 6.0 years and 32.8 ± 7.1 kg for males and 72.7 ± 6.0 years and 21.6 ± 4.8 kg for females, respectively. Additionally, the most predominant chronic disease in both sexes was hypertension (male: 43.2%; female: 44.5%) followed by DM (male: 20.9%; female: 19.8%).

The distribution (mean ± SD) of age, BMI, and GS in different chronic diseases among males (n = 1600) are presented in Table 2. Participants with hyperlipidaemia had the highest BMI, while those without hypertension had the lowest. Participants without CVD and CVA demonstrated the highest GS, whereas those with CVA had the lowest. Among these seven specified chronic diseases, CVD, CVA, and CKD revealed a statistically significant decrease in GS (*p* < 0.05, *p* < 0.001, and *p* < 0.01, respectively), while hypertension, hyperlipidaemia, DM, and all cancer types demonstrated no statistical significance. GS decreased the most among participants with CVA. Moreover, GS significantly decreased as the number of diseases statuses increased (*p* < 0.05).

Table 3 presents the distribution (mean ± SD) of age, BMI, and GS in different chronic diseases among females (n = 2139). Participants with CVA had the highest BMI, while those without hypertension had the lowest. Those participants with cancer (all types) had the highest GS, whereas those with DM had the lowest. Among these seven specified chronic diseases, cancer of all types revealed a statistically significant increase in GS (*p* < 0.01), while CVD, hypertension, and DM demonstrated a statistically significant decrease (*p* < 0.05, *p* < 0.01 and *p* < 0.01, respectively). In addition, GS demonstrated no statistically significant change in participants with CVA, hyperlipidaemia, and CKD. Furthermore, GS significantly decreased as the number of diseases increased (*p* < 0.05).

The multiple regression coefficients for GS on chronic disease statuses, before and after adjusting for the potential confounding factors with sex specification, are demonstrated in Table 4. In the crude model (Model I), there was a statistically significant inverse association with participants who had two diseases (the coefficient was −1.25 with *p* = 0.014 in males and −0.78 with *p* = 0.008 in females) or more than three diseases (the coefficient was −1.39 with *p* = 0.030 in males and −0.76 with *p* = 0.056 in females) in both sexes. However, it was only statistically significant in male participants with one disease (the coefficient was −0.84 with *p* = 0.045), not females. After adjusting for age and BMI, the regression coefficient (Model II) indicated a borderline inverse association when the participants had two diseases (the coefficient was −0.89 with *p* = 0.051 in males and −0.51 with *p* = 0.068 in females) or more than three diseases in males (the coefficient was −1.38 with *p* = 0.017). However, no statistical significance was observed when the participants had only one disease in both sexes, and more than three diseases in females.

Table 5 summarizes the distribution of GS among different chronic diseases statuses and BMI statuses with sex specification. We divided the participants into three subgroups, namely, BMI I (<24 kg/m^2^), BMI II (24–27 kg/m^2^), and BMI III (≥27 kg/m^2^), to discuss the association between GS and the number of chronic diseases among these subgroups. In both sexes, obese (BMI III) had the highest GS, while normal weight (BMI I) had the lowest. Furthermore, GS did not significantly change as the disease number increased in the BMI III subgroup, in both sexes (*p* = 0.117 in males and *p* = 0.630 in females). However, GS significantly decreased as the disease number increased in the BMI II subgroup for both sexes (*p* = 0.009 in males and *p* = 0.026 in females). In the BMI I subgroup, GS demonstrated either a borderline significant association or no significant association at all (*p* = 0.211 in males and *p* = 0.042 in females).

Figure 1 demonstrates the minimum, maximum, median, and interquartile range of grip strength among different disease statuses in the elderly, for both sexes. There is a slight decrease in GS in those who had two chronic diseases.

## 4. Discussion

With a rising ageing population, the health of older adults has become an important issue not only in health systems but also as an economic burden in recent years. GS is not only an objective marker but also a convenient tool to predict one’s functional status, nutritional status, cognitive function, and activities of daily living (ADL) [20]. Lower GS was inversely associated with all-cause mortality [21] but was especially associated with cardiovascular and respiratory diseases and cancer [22].

Our study found that GS was inversely associated with multimorbidity among community-dwelling people over age 65 in Taiwan, consistent with previous studies [12,14,15,17]. However, no statistical significance was observed when age and BMI were further adjusted in females. Nonetheless, the consistency of inverse relationships between GS and multimorbidity in older adults of both sexes remains reliable.

This study has some strengths. First, the number of participants was large, which strengthens the analyses of association. We also discussed the association between each specific chronic disease and GS. Furthermore, we investigated the relationships of GS and multimorbidity with three weight statuses (normal weight, overweight, and obesity). This study is the first to investigate such association among an Asian population.

The negative association between sarcopenia and CVD has an evidence-based consensus [23]. Furthermore, a bidirectional association was reported, indicating that sarcopenia may lead to CVD and CVD may induce sarcopenia [23]. Our study also presented a consistent result, that is, GS was significantly reduced in participants with CVD in both sexes.

Not surprisingly, GS decreased the most in older male participants with CVA. In contrast, GS did not significantly change in females with CVA. However, in a previous study, females had a poorer outcome of activity limitation (assessed from the modified Rankin Scale) after having a stroke when compared to males [24], despite the relatively low stroke incidence adjusted by age [25]. Our study may also have healthcare participant bias because patients with severe stroke were usually bedridden, preventing them from participating in our investigation.

As age increases, people with diabetes mellitus exhibit a more rapid loss of function and muscle mass [26]. Many bidirectional relationships exist between diabetes and sarcopenia [27], though the true pathophysiology remains unclear. The possible mechanism included diabetes-related oxidative stress [28] and inflammation [29], accumulation of advanced glycation end products [30], and increased insulin resistance [31]. Our study also indicated that the GS trend declined in older males with diabetes mellitus but significantly reduced in females with diabetes mellitus individually. No statistical significance was observed in males, possibly because of limited questionnaire-related information (such as the duration and control status of diabetes); nevertheless, the inverse association between GS and diabetes mellitus remains consistent with previous studies [12,17,32].

Chronic kidney disease is highly associated with sarcopenia because of catabolic state–related protein wasting, inflammation-related dysregulation of myostatin and activation of the ubiquitin–proteasome system (caspase-3) [33]. In our study, GS significantly reduced only in males with CKD. This sex difference is consistent with previously published studies [34,35], but the true mechanism is still unclear. Further large-scale studies focusing on the possible pathophysiology should be conducted.

Surprisingly, cancer was not associated with GS in males and demonstrated a significantly positive association in females in our study. This result is similar to a published large-scale study (Prospective Urban–Rural Epidemiology study), which reported that low muscle strength is associated with a low risk for cancer [36]. However, a reasonable explanation remains unknown and may also be associated with participant bias. Hence, further studies should be conducted, including discussion on the different types of cancer.

Identifying the ideal body weight for older adults is extremely challenging and remains debatable. Older people with suitable excess weight may have a lower risk than younger people [37,38]. Giovanni et al. recently conducted a large cross-sectional study involving 3455 older individuals (over age 60) in Italy. Interestingly, their study results revealed that older individuals with a higher BMI of 27.5–29.9 kg/m^2^ (similar with BMI III in our study) had a lower relative risk of developing comorbidity than those with a BMI of 25.0–27.4 kg/m^2^ (similar with BMI II) [39]. Our study also indicated similar results (Table 5) wherein the older participants with a BMI beyond 27 kg/m^2^ had a higher GS than those with BMIs below 24 kg/m^2^ and 24.0–27.0 kg/m^2^ in both sexes. More interestingly, GS indicated no difference even when the number of chronic diseases increased in older participants with a BMI of 27 kg/m^2^ or higher, in both sexes. These results could be justified by the notion that older people with mild to moderate excess weight did not experience malnutrition (abnormally low BMI), which can generally result in tooth loss, depression, and severe illnesses. Hence, suitable excess weight may be a protective factor in older adults; it may indicate good nutritional status in older adults even with multimorbidity. However, further large-scale studies combining GS and appendicular muscle mass with validated equations [40], and the percentage of fat and muscle are needed to prove and emphasize this hypothesis. 

This study also has some limitations. Firstly, false negatives are possible because of the likelihood of including undiagnosed diseases and the bias of questionnaires and questioning. Secondly, although we used the seven most frequent older adult diseases, we ignored the potential impact of nutritional status and psychological disorders, such as depression and anxiety. However, the relationship between GS and multimorbidity among older populations with different weight statuses demonstrated in this study is still reliable.

## 5. Conclusions

In this study, grip strength was inversely associated with multimorbidity status in the elderly, implying that we should consider the possibility of sarcopenia and muscle weakness each time we assess an older adult with multiple chronic diseases, especially CVD and CVA. Our results also support that body weight status and grip strength may be important indicators of health status in older adults. Body weight and grip strength should be measured during follow-ups in older adults to prevent the development of complications from such chronic diseases in later life.

## Figures and Tables

**Figure 1 ijerph-18-07540-f001:**
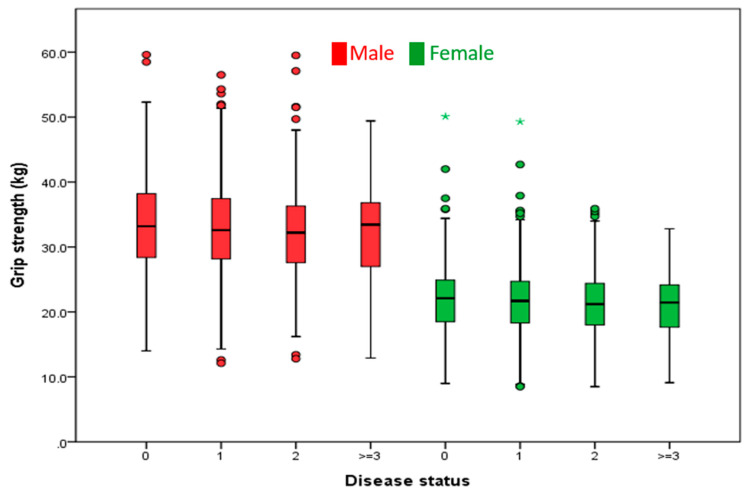
Box plots of grip strength among different disease statuses in the elderly.

**Table 1 ijerph-18-07540-t001:** Distributions (mean ± SD) of general characteristics and chronic disease status among study population (n = 3739).

	Total	Males	Females
Variables	(n = 3739)	(n = 1600)	(n = 2139)
	Mean ± SD	Mean ± SD	Mean ± SD
Age (years)	72.9 ± 6.0	73.2 ± 6.1	72.7 ± 6.0
Body height (cm)	155.8 ± 8.2	162.4 ± 6.0	150.9 ± 5.7
Body weight (kg)	60.9 ± 10.6	65.8 ± 10.1	57.1 ± 9.4
BMI (kg/m^2^)	25.0 ± 3.7	24.9 ± 3.4	25.1 ± 3.8
Grip strength (kg)	26.4 ± 8.1	32.8 ± 7.1	21.6 ± 4.8
Chronic disease (%)			
CVD	15.4%	15.8%	15.1%
CVA	2.6%	3.9%	1.7%
Hypertension	43.9%	43.2%	44.5%
Hyperlipidemia	14.9%	13.7%	15.8%
DM	20.2%	20.9%	19.8%
All cancer	2.8%	3.3%	2.5%
CKD	3.1%	4.1%	2.4%
Disease status (%)			
0 disease	36.7%	37.2%	36.4%
1 disease	35.6%	34.5%	36.5%
2 diseases	18.6%	18.6%	18.6%
≥3 diseases	9.2%	9.7%	8.6%

Abbreviations: BMI, body mass index; CVD, cardiovascular disease; CVA, cerebrovascular disease; DM, diabetes mellitus; CKD, chronic kidney disease.

**Table 2 ijerph-18-07540-t002:** Age, BMI, and grip strength distributions (mean ± SD) in different chronic disease statuses among male participants (n = 1600).

	Age		BMI		GS	
Variables	Mean ± SD		Mean ± SD		Mean ± SD	
CVD ^$^						
No	73.0 ± 6.0	***	24.8 ± 3.4	**	33.0 ± 7.1	*
Yes	74.5 ± 6.4		25.5 ± 3.6		31.8 ± 7.2	
CVA						
No	73.2 ± 6.1		24.9 ± 3.4		33.0 ± 7.1	***
Yes	74.3 ± 5.7		25.0 ± 3.0		29.5 ± 7.3	
Hypertension						
No	72.6 ± 5.9	***	24.3 ± 3.2	***	33.0 ± 7.1	
Yes	74.1 ± 6.1		25.8 ± 3.5		32.6 ± 7.1	
Hyperlipidemia						
No	73.3 ± 6.1		24.8 ± 3.4	***	32.8 ± 7.1	
Yes	72.6 ± 5.6		26.0 ± 3.4		33.1 ± 7.2	
DM						
No	73.4 ± 6.1		24.8 ± 3.4	***	33.0 ± 7.2	
Yes	72.8 ± 6.1		25.6 ± 3.5		32.2 ± 6.8	
All cancer						
No	73.2 ± 6.1		24.9 ± 3.4		32.8 ± 7.2	
Yes	74.1 ± 6.0		24.8 ± 3.2		32.1 ± 6.7	
CKD						
No	73.1 ± 6.0	***	24.9 ± 3.4		32.9 ± 7.1	**
Yes	76.0 ± 5.8		24.9 ± 3.6		30.4 ± 7.4	
Disease status ^†^						
0 disease	72.6 ± 6.0	**	24.1 ± 3.1	***	33.5 ± 7.3	*
1 disease	73.4 ± 5.9		25.1 ± 3.4		32.6 ± 7.0	
2 diseases	74.0 ± 6.3		25.4 ± 3.6		32.2 ± 6.9	
≥3 diseases	73.8 ± 6.1		26.5 ± 3.3		32.1 ± 7.3	

Abbreviations: CVD, cardiovascular disease; CVA, cerebrovascular disease; DM, diabetes mellitus; CKD, chronic kidney disease. ^$^ Compared between with and without disease using the student *t*-test: * *p* < 0.05, ** *p* < 0.01, *** *p* < 0.001. ^†^ Compared among disease statuses using the ANOVA test: * *p* < 0.05, ** *p* < 0.01, *** *p* < 0.001.

**Table 3 ijerph-18-07540-t003:** Age, BMI, and grip strength distributions (mean ± SD) in different chronic disease statuses among female participants (n = 2139).

	Age		BMI		GS	
Variables	Mean ± SD		Mean ± SD		Mean ± SD	
CVD ^$^						
No	72.5 ± 5.9	***	25.0 ± 3.8		21.7 ± 4.8	*
Yes	74.1 ± 6.1		25.4 ± 3.9		21.1 ± 4.9	
CVA						
No	72.7 ± 6.0		25.1 ± 3.8	*	21.6 ± 4.8	
Yes	73.6 ± 5.3		26.4 ± 4.1		21.6 ± 4.4	
Hypertension						
No	72.0 ± 5.7	***	24.4 ± 3.6	***	21.9 ± 4.9	**
Yes	73.6 ± 6.1		26.0 ± 4.0		21.3 ± 4.7	
Hyperlipidemia						
No	72.7 ± 6.0		25.0 ± 3.9	*	21.6 ± 4.8	
Yes	72.7 ± 6.0		25.5 ± 3.8		21.8 ± 5.1	
DM						
No	72.7 ± 6.0		25.0 ± 3.8	**	21.8 ± 4.9	**
Yes	72.8 ± 5.7		25.6 ± 3.9		21.0 ± 4.5	
All cancer						
No	72.7 ± 6.0		25.1 ± 3.8		21.6 ± 4.8	*
Yes	71.4 ± 4.7		25.2 ± 4.2		23.1 ± 5.7	
CKD						
No	72.7 ± 5.9	**	25.1 ± 3.9		21.6 ± 4.8	
Yes	74.7 ± 6.2		24.9 ± 3.7		21.4 ± 5.2	
Disease status ^†^						
0 disease	71.8 ± 5.9	***	24.3 ±3.5	***	21.9 ± 4.8	*
1 disease	73.0 ± 5.9		25.3 ± 4.0		21.6 ± 4.9	
2 diseases	73.5 ± 5.9		25.9 ± 3.9		21.2 ± 4.8	
≥3 diseases	73.6 ± 6.3		25.9 ± 3.9		21.2 ± 4.7	

Abbreviations: CVD, cardiovascular disease; CVA, cerebrovascular disease; DM, diabetes mellitus; CKD, chronic kidney disease. ^$^ Compared between with and without disease using student *t*-tests: * *p* < 0.05, ** *p* < 0.01, *** *p* < 0.001. ^†^ Compared among disease statuses using the ANOVA test: * *p* < 0.05, *** *p* < 0.001.

**Table 4 ijerph-18-07540-t004:** Multivariate regression coefficients of grip strength on chronic disease status before and after adjusting for confounding variables with sex specifications.

	Model I ^†^			Model II ^‡^		
	β	se β	*p*-Value	β	se β	*p*-Value
Men(n = 1600)						
1 disease/0 disease	−0.84	0.42	0.045	−0.66	0.38	0.077
2 diseases/0 disease	−1.25	0.51	0.014	−0.89	0.45	0.051
≥3 diseases/0 disease	−1.39	0.64	0.030	−1.38	0.58	0.017
Women(n = 2139)						
1 disease/0 disease	−0.32	0.24	0.193	−0.11	0.23	0.635
2 diseases/0 disease	−0.78	0.30	0.008	−0.51	0.28	0.068
≥3 diseases/0 disease	−0.76	0.39	0.056	−0.46	0.37	0.216

Abbreviations: β, regression coefficient; se, standard error. ^†^ Model I: no adjustment. ^‡^ Model II: further adjusting for age and body mass index.

**Table 5 ijerph-18-07540-t005:** Distributions (mean ± SD) of grip strength among different chronic diseases statuses and BMI status with sex specification (n = 3739).

	Grip Strength			
BMI Status	Males (n = 1600)		Females (n = 2139)	
	Mean ± SD		Mean ± SD	
BMI I (<24)	(n = 643)		(n = 897)	
0 disease	32.0 ± 7.1		21.3 ± 4.4	
1 disease	31.3 ± 6.5		20.7 ± 4.5	
2 diseases	30.9 ± 6.0		20.3 ± 4.4	
≥3 diseases	29.9 ± 7.7		20.0 ± 4.5	
	F = 1.507	*p* = 0.211 ^†^	F = 2.742	*p* = 0.042 ^†^
BMI II (24–27)	(n = 560)		(n = 645)	
0 disease	34.4 ± 7.2		22.4 ± 4.9	
1 disease	33.1 ± 6.9		22.1 ± 4.9	
2 diseases	32.0 ± 6.6		21.0 ± 4.9	
≥3 diseases	31.7 ± 7.3		21.1 ± 4.2	
	F = 3.881	*p* = 0.009 ^†^	F = 3.096	*p* = 0.026 ^†^
BMI III (≥27)	(n = 397)		(n = 597)	
0 disease	35.9 ± 7.4		22.9 ± 5.2	
1 disease	34.0 ± 7.3		22.4 ± 5.1	
2 diseases	34.2 ± 8.0		22.1 ± 5.0	
≥3 diseases	33.4 ± 7.0		22.3 ± 5.1	
	F = 1.979	*p* = 0.117 ^†^	F = 0.578	*p* = 0.630 ^†^

Abbreviations: BMI, body mass index (kg/m^2^); BMI I, BMI < 24 kg/m^2^; BMI II, BMI: 24–27 kg/m^2^; BMI III, BMI ≥ 27 kg/m^2^. ^†^ Compared among disease status in the same sex and BMI status using the ANOVA test.

## Data Availability

Please contact correspondence author to access available data.

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
