# Peer review of "Relationship of Multimorbidity, Obesity Status, and Grip Strength among Older Adults in Taiwan"

_ijerph, 2021, doi:10.3390/ijerph18147540_

Round 1

Reviewer 1 Report

Congratulations on your manuscript. You present an investigation with a very large population sample, which provides validity to the study. I will make a consideration.

  • Line 43: I recommend that you delve into this idea. You must consider not only changes in the muscular system but also in the nervous system. I recommend that you consult the following manuscript: Series A: Biomedical Sciences Medical Sciences, 67(1), 28-40. https://doi.org/10.1093/gerona/glr010 
  • Line 45: You indicates that the GS allows determining the nutritional status. However, I think you should explain that additional tests it would be needed to accurately determine the nutritional status of the person evaluated.
  • In your objectives I think you should make it clear what is the applicability of your study.
  • Justify why they did not use a bioimpedance Tanita (to be able to determine the percentage of fat and muscle) or waist-hip measurements to assess cardiovascular risk.
  • I recommend that you indicate whether the participants signed an informed consent to participate as well as all the information related to the study.
  • In the discussion, you state that an adequate excess weight could be ideal for the elderly. I agree with you since different studies support this idea and some even propose a BMI of between 25 to 27. However, and even though you indicate that more studies are required to corroborate this fact, I think you should emphasize this idea, since that both the fat and muscle percentage of the person evaluated should be considered.
  • The limitations of the study should be stated at the end of the discussion.

Reviewer 2 Report

  1. This manuscript adds to the growing literature on the importance of sarcopenia in predicting the health status of older adults.
  2. The manuscript is scientifically sound and well written.
  3. Text recommendations: Delete sentence on lines 251 - 254 as speculative. Delete the last sentence beginning line 260-263.  Consider rephrasing: Body weight and grip strength may be important indicators of health status in older adults.
  4. Grammar changes: line 17- 65 years-old; line 34-diseases and is an important...; line 38- increased; line 42-systems; line 51 - is also important; line 75-using standard methods; line 146 - are demonstrated; line 195 - false negatives are; line 198 - most frequent older adult diseases; line 199 - nutritional status and psychological disorders; line 237 - people with higher BMI; line 251 - in older adults.
  5. Consider including the questionnaire in an appendix

Reviewer 3 Report

In this study, the authors evaluated the relationship between handgrip strength (HGS) and the presence of one or more chronic pathologies (CVD, HT, CVA, cancer, DM, CKD) and risk factors (hyperlipidemia) and the covariant effect of body fatness and sex in 3,739 Taiwanese elders (73 ± 6 y, 57.2 female). Using an unconventional statistical approach, the study shows that BMI> age> HGS independently predict pluripatology in a gender-dependent manner; GS alone predicted CVA> CKD> CVD (men), HT> DM> CVD (women), and having two diseases (p≤ 0.01) but not one or ≥3 (p>0.05). The study presents relevant information, but some modifications are required to improve the scientific soundness and scientific contribution:

General comments

  • The readability and syntax of the manuscript will be substantially improved if it is reviewed by a formal translation agency or by a native English spoken person.
  • In this study, GS cut off points were not used to stratify your population into "poor" or "good" GS (doi: 10.6288/TJPH.201812_37(6).107061) and instead you used individual GS values to estimate the means and standard deviations for each studied factor, missing a great opportunity to analyze logistic regression trends and specific GS cut of values based on ROC analysis. The authors should explore this alternative of data presentation, considering the following studies as examples (doi): 10.1111/j.1532-5415.2010. 03035.x, 10.4236/jbm.2014.29003, 10.1016/j.clnu.2016.02.002, 10.1016/j.clnu.2018.03.012.
  • The important covariant effect of sex, BMI, and age on the predictive value of GS for mono/pluripathology should be explored by multinomial regression analysis.
  • To make the study findings more meaningful, the authors are advised to calculate the appendicular muscle mass with the recently validated equation for the Taiwanese population (https://doi.org/10.1016/j.jamda.2020.08.003), foreseeing that the authors have all the information for its calculation and could propose it as another predictor parameter of pluripathology. 

Title. OK.   

Abstract. It should be more concise without sacrificing important differential results, highlighting. For example: “GS alone predicted CVA> CKD> CVD (men), HT> DM> CVD (women) and having two diseases (p≤ 0.01) but not one or ≥3 (p>0.05).”.

Introduction & conclusion.  The authors should give more emphasis to the value of GS as clinical morbidity and all/specific mortality in the elderly (doi: https://www.ncbi.nlm.nih.gov/pmc/articles/PMC6778477/), as well as its use in longitudinal and cross-sectional epidemiological studies in Taiwan.

Methods.

  • Lines 66-73: It is particularly important for this study that it be argued with previous studies on the validity of the self-report of diseases described in the section (questionnaire).
  • Line 84: “was measured with a digital hand dynamometer with adjustable grip (TKK 5101Grip D; Takey, Tokio Japan), following the procedure described by España-Romero et al. (20). Briefly, …”. Please include range and precision.
  • Lines 93-102: If it is decided to include correlation tests, they should be described in the statistics section.

Results & discussion. This section should change according to the other suggested changes.

Tables (T) & Figures (F). The authors must include graphs showing the linear (logistic) trend between the total number of pathologies or cumulative prevalence of specific ones (particularly CVA) with the specific value of GS. See the following example (DOI): 10.1038 / s41598-020-63713-1. Remember that figures should be provided with high resolution (≥300 dpi).

References. Authors should reduce the number of references ≥10y old to say 30% [currently 43% (19/44)].

Round 2

Reviewer 3 Report

Thank you for accepting some of my comments.